# Factors Limiting the Apparent Hydrogen Flux in Asymmetric Tubular Cercer Membranes Based on La_27_W_3.5_Mo_1.5_O_55.5−*δ*_ and La_0.87_Sr_0.13_CrO_3−*δ*_

**DOI:** 10.3390/membranes9100126

**Published:** 2019-09-24

**Authors:** Zuoan Li, Jonathan M. Polfus, Wen Xing, Christelle Denonville, Marie-Laure Fontaine, Rune Bredesen

**Affiliations:** SINTEF Industry, Thin Film and Membrane Technology, P.O. Box 124 Blindern, NO-0314 Oslo, Norway; Jonathan.Polfus@sintef.no (J.M.P.); wen.xing@sintef.no (W.X.); Christelle.Denonville@sintef.no (C.D.); Marie-Laure.Fontaine@sintef.no (M.-L.F.); Rune.Bredesen@sintef.no (R.B.)

**Keywords:** hydrogen permeation, water splitting, surface kinetics, asymmetric tubular membrane, lanthanum tungstate, lanthanum chromite

## Abstract

Asymmetric tubular ceramic–ceramic (cercer) membranes based on La_27_W_3.5_Mo_1.5_O_55.5−*δ*_-La_0.87_Sr_0.13_CrO_3−*δ*_ were fabricated by a two-step firing method making use of water-based extrusion and dip-coating. The performance of the membranes was characterized by measuring the hydrogen permeation flux and water splitting with dry and wet sweep gases, respectively. To explore the limiting factors for hydrogen and oxygen transport in the asymmetric membrane architecture, the effect of different gas flows and switching the feed and sweep sides of the membrane on the apparent hydrogen permeability was investigated. A dusty gas model was used to simulate the gas gradient inside the porous support, which was combined with Wagner diffusion calculations of the dense membrane layer to assess the overall transport across the asymmetric membrane. In addition, the stability of the membrane was investigated by means of flux measurements over a period of 400 h.

## 1. Introduction

Technology that can efficiently separate hydrogen from a mixed gas stream may be integrated in processes that either consume or liberate hydrogen (e.g., in hydrocarbon upgrading such as non-oxidative methane aromatization) or employed in stand-alone hydrogen production [1,2,3]. Dense metallic membranes based on Pd and its alloys show high solubility and diffusivity of hydrogen and are promising for hydrogen separation at intermediate temperatures (>~300–500 °C) [4,5]. On the other hand, dense ceramic membranes based on mixed proton and electron conducting materials demonstrate thermal and chemical stability for application in high temperature processes (>700 °C) [6,7], although their hydrogen fluxes are much lower than their metallic counterparts.

Mixed proton and electron conducting membranes based on perovskite-type materials, such as acceptor-doped SrCeO_3_ and BaCeO_3_, suffer either poor stability, e.g., in CO_2_-containing atmospheres, or low hydrogen flux as a consequence of low electronic conductivity under relevant operational conditions [6,8,9,10,11]. Rare earth tungstates have attracted much attention due to their high chemical stability in CO_2_ and H_2_S containing environments [12,13,14]. Molybdenum substituted rare-earth tungstates, La_27_W_3.5_Mo_1.5_O_55.5−*δ*_ (LWM), show relatively high hydrogen permeability [15,16] and chemical stability in the presence of CO_2_ [17]. To further increase the electronic conductivity of LWM, composite ceramic–ceramic (cercer) membranes including acceptor-doped La_0.87_Sr_0.13_CrO_3−*δ*_ (LSC) have been extensively studied [18,19,20]. Proton transport in these membranes mainly proceeds through the LW(M) phase, while the LSC phase contributes with high electronic conductivity. The improved performance of the cercer membranes as compared to the single-phase LW(M) becomes more significant at lower temperatures, i.e., 500–750 °C, where the apparent H_2_ permeability of LW(M) is limited by the electronic conductivity [21]. Furthermore, water splitting at the sweep side and subsequent oxide ion transport through the membrane was found to predominate the apparent hydrogen permeability (defined as the rate of hydrogen transported across the membrane plus hydrogen produced by water splitting on the sweep side) at higher temperatures with H_2_O-containing sweep gas [22]. Apparent H_2_ permeabilities of 1–5 × 10^−3^ mL min^−1^ cm^−1^ have been obtained at 700 °C with around 50% H_2_ (humidified) feed and humidified Ar sweep at ambient total pressure [18,19]. This apparent permeability was, however, significantly enhanced by coating the 1-mm thick disc shaped cercer membranes with a porous Pt-layer, indicating that surface kinetics can significantly impact the apparent flux. Flux degradation has been observed at temperatures above 900 °C and was related to equilibration of the La/W ratio in the LWM phase under reducing conditions, resulting in lower ionic conductivity of the LW(M) phase and consequently reduced flux [23].

An asymmetric tubular membrane design has been considered promising for increasing flux when the membrane layer is thicker than the characteristic thickness or catalyzed on the surfaces [13,24,25]. Moreover, a tubular membrane geometry may offer better flow profiles with less risk of dead zones, better thermal cycling stability, and easiness to seal in comparison with a planar design [26,27]. 

In this investigation, we evaluate the performance of asymmetric tubular membranes consisting of a dense LWM-LSC cercer layer supported by a porous LWM support in terms of hydrogen flux as a function of temperature using both wet and dry sweep gases. To explore the limiting factors for transport through the membrane and the support, various experiments were conducted including variation of the sweep flow rate and switching the feed and sweep side of the membrane. Furthermore, numerical simulations of mass transport across the asymmetric membranes were carried out by means of a dusty gas model in the porous support and Wagner ambipolar transport in the membrane layer.

## 2. Materials and Methods 

### 2.1. Fabrication of Asymmetric LWM–LSC Tubular Membranes

La_27_W_3.5_Mo_1.5_O_55.5−*δ*_ (LWM) and La_0.87_Sr_0.13_CrO_3−δ_ (LSC) powders were purchased from Marion Technology (Verniolle, France) and Praxair (Indianapolis, IN, USA), respectively. Batches of LWM feedstock were prepared for extrusion of the porous tubular supports by mixing LWM powder with pore formers and a proprietary water soluble binder system. Extrusion was carried out in a Loomis 40-ton ram extruder with a ram speed of 1 cm/s to obtain straight tubes of 40–50 cm length, which were further cut into pieces of 20–30 cm length and bisque-fired in air at 1300 °C for 5 h. A suspension of LWM-LSC (70/30 wt.%) was used for dip-coating the bisque-fired tubes using a semi-automatic dip-coater. After dipping, the membranes were sintered in air for 10 h. Further details regarding the fabrication procedure can be found elsewhere [27]. The membrane used for flux measurements was 5.4 cm long with an outer diameter of 1 cm and 50 µm thickness for the dense layer. In order to catalyze the surface kinetics, a porous layer of Pt was applied by painting a Pt ink (Englehart) layer on the outer side of the membrane.

### 2.2. Hydrogen Permeation Measurements

The tubular membranes were bubble tested at room temperature in isopropanol up to 3 bar to ensure leak-free membranes prior to flux measurements. The asymmetric tubular membrane was closed in one end using a dense ZrO_2_ cap, and the other end was sealed to a ZrO_2_ support tube (outer diameter 13 mm) using an alumina paste (671 Ceramabond, Aremco, Valley Cottage, NY, USA). A proprietary glass–ceramic seal was subsequently applied on all joints. The setup was mounted into a ProboStat measurement cell (Norecs AS, Oslo, Norway) enclosed with an outer quartz tube, and heated to 1100 °C with a heating rate of 10 °C/h to obtain a sufficient seal. The temperature was monitored with an S-type thermocouple positioned close to the center of the tubular membrane. The feed side of the membrane (outer) was fed with a mixture of 25 mL/min He and 25 mL/min H_2_, and the permeate side (inner) was swept with Ar (25 mL/min). The gas mixtures were humidified with about 2.5% H_2_O by passing the gas through a saturated KBr solution. The sealing process was monitored by measuring the He and H_2_ concentrations in the permeate side during heating using a Varian CP-4900 gas chromatograph (GC) (Palo Alto, CA, USA). For the investigated membranes, the helium leakage was ~0.0005% of the total H_2_ concentration in the sweep throughout the experimental window and leakage correction was therefore not required for calculating the apparent fluxes.

Apparent hydrogen fluxes were measured from 1100 to 700 °C with feed gas mixtures of 48.75% H_2_, 2.5% H_2_O, and 48.75% He using both dry and wet Ar as sweep gas with heating and cooling rates of 10 °C/h. The effect of sweep flow on the apparent hydrogen flux was studied by varying the Ar sweep flow rate from 25 to 200 mL/min. The total pressure was atmospheric for both the feed and sweep side during all measurements. To study the potential effects of mass transport resistance in the porous support and surface kinetics at the feed side, the feed and sweep sides of the membranes were exchanged and various sweep flow rates were applied. In addition, the stability of the tubular membrane was verified by measuring the flux at 750 °C for 400 h.

### 2.3. Numerical Modeling for Gas Transport in Porous Support

To understand the effects of a porous support on the hydrogen flux of an asymmetric membrane, it is necessary to know the hydrogen and steam partial pressure gradients inside the support, which in turn determine the driving force for hydrogen and oxygen permeation through the dense membrane layer. Since the thickness of the support is far smaller than its length, the gradient is mainly in the radial direction. Therefore, the transport through the porous media is considered one-dimensional. The species and overall mass-conservation equations are expressed as:(1)$$\frac{\partial \left(\phi_{g}\rho_{g}Y_{k}\right)}{\partial t}+\nabla \cdot j_{k}=0$$
(2)$$\frac{\partial \left(\phi_{g}\rho_{g}\right)}{\partial t}+\overset{K_{g}}{\underset{k=1}{\sum }}\nabla \cdot j_{k}=0$$
where $\rho_{g}$ is the gas-phase density, $Y_{k}$ is the mass fraction, and $\phi_{g}$ is the porosity of the support. The gas-phase density is determined from the equation of state as:(3)$$\rho_{g}=\frac{p}{RT_{g}\sum_{k=1}^{K_{g}}\frac{Y_{k}}{W_{k}}}$$
where $W_{k}$ is the molar mass. The mass fluxes $j_{k}$ of species *k* are related to molar fluxes $J_{k}$ through the Dusty-gas model (DGM) [28,29] according to
(4)$$\underset{l\neq k}{\sum }\frac{\left[X_{l}\right]J_{k}-\left[X_{k}\right]J_{l}}{\left[X_{T}\right]D_{kl}^{e}}+\frac{J_{k}}{D_{k,\mathrm{Kn}}^{e}}=-\nabla \left[X_{k}\right]-\frac{\left[X_{k}\right]}{D_{k,\mathrm{Kn}}^{e}}\frac{B_{g}}{\mu }\nabla p$$
where $\left[X_{k}\right]$ is the molar concentration, $\left[X_{T}\right]=p/RT$ the total molar concentration, $B_{g}$ the permeability, and μ the mixture viscosity. $D_{kl}^{e}$ and $D_{k,\mathrm{Kn}}^{e}$ are the effective ordinary and Knudsen diffusion coefficients respectively, can be evaluated as
(5)$$D_{kl}^{e}=\frac{\phi_{g}}{\tau }D_{kl},D_{k,\mathrm{Kn}}^{e}=\frac{4}{3}\frac{r_{p}\phi_{g}}{\tau }\sqrt{\frac{8RT}{\pi W_{k}}}$$

The binary diffusion coefficients $D_{kl}$ are calculated from kinetic theory. Knudsen diffusion represents mass transport assisted by gas–wall collisions, and depends on the average pore radius $r_{p}$ and the tortuosity τ.

The system of equations forms an initial-boundary-value problem in differential-algebraic form, and boundary conditions are required for both sides of the support. At the membrane/support interface, the hydrogen and steam flux are specified according to the measured flux, and the measured gas compositions of the sweep gas are used as gas composition at the support-gas interface at the inner tube side. The model was implemented to solve the transient problem using one-dimensional finite-volume spatial discretization along the radial direction of the support by means of the ode15i function in the MATLAB software package (R2016a).

## 3. Results and Discussion

### 3.1. Characterization of Tubular Membranes 

The SEM images of the tubular membranes after sintering at 1500 °C show a dense LWM-LSC membrane layer and a porous support of LWM exhibiting a ‘worm-like’ porous network (c.f. Figure 1). The LSC phase is embedded in the LWM phase and uniformly distributed in the dense membrane layer. A closer inspection of the membrane cross-section highlights another critical feature: the interface between the membrane layer and porous support consists of a quite dense region of 1–2 microns which appears to contain no LSC grains. This microstructure results from the processing of the membranes and the large difference between the sinterability of the LSC and LWM phases (c.f. [27]). We will discuss further how this microstructure impacts on the performance of the membranes.

The porous LWM support sintered with the same annealing program showed a permeability above 1 × 10^−^^14^ m^2^ and the support should thereby not be resistant for overall gas diffusion [27]. The porosity of the tubular supports was determined from mercury intrusion porosimetry (Autopore IV 9500, Norcross, USA) and tortuosity was assumed to be 3. Table 1 lists various parameters of the porous support used for modeling.

### 3.2. Hydrogen Permeation

Figure 2 presents the measured apparent hydrogen flux as a function of the inverse absolute temperature during cooling for various feed and sweep gases: wet feed + wet sweep (denoted WF+WS), wet feed + dry sweep (WF + DS), and dry feed + dry sweep (DF + DS). It is obvious from Figure 2 that the highest hydrogen flux was obtained for WF + WS, while the lowest was obtained for DF + DS. 

For a bulk-controlled hydrogen transport, the hydrogen flux across a membrane in a hydrogen potential gradient can be related to the ambipolar transport of protons and electrons [6]
(6)$$j_{H_{2}}=-\frac{RT}{4F^{2}L}\int_{{\mathrm{pH}}_{2}^{I}}^{{\mathrm{pH}}_{2}^{\mathrm{II}}}\sigma_{H^{+}}t_{e^{-}}d\left(\ln pH_{2}\right)$$
where $\sigma_{H^{+}}$ and $t_{e^{-}}$ are the proton conductivity and electronic transport number respectively, ${\mathrm{pH}}_{2}^{\mathrm{II}}$ and ${\mathrm{pH}}_{2}^{I}$ are the hydrogen partial pressures at the sweep and feed side, respectively, *L* the membrane thickness, and the remaining symbols have their usual meanings. 

Protons may dissolve into the membrane material by hydration of oxygen vacancies via
(7)$$H_{2}O\left(g\right)+V_{O}^{••}+O_{O}^{\times }={2\mathrm{OH}}_{O}^{•}$$

The above reaction is exothermic, and as temperature increases the proton concentration decreases while mobility increases, which in total leads to a maximum and subsequently an overall reduction in proton conductivity. According to Equation (6), the hydrogen flux will also become less dependent on temperature with dry sweep. In the case of WF + DS in Figure 2, the hydrogen flux shows a lower temperature dependence at high temperatures, indicating a proton-limited transport, while a higher dependence at low temperatures corresponds to electron-limited transport. Under both dry feed and sweep, hydrogen permeation is still observed. In this case, protons may dissolve in oxides with simultaneous formation of electrons according to the following defect reaction:(8)$$H_{2}\left(g\right)+2O_{O}^{\times }={2\mathrm{OH}}_{O}^{•}+2e^{/}$$

Under dry conditions, it seems that the kinetics for the above reaction are very slow as indicated by the increase in hydrogen flux with dwell time at 850 and 750 °C. In this case, it is hard to determine the limiting factors for hydrogen permeation in the whole temperature range. 

With wet sweep gas, the apparent hydrogen flux was higher than for dry sweep in the measured experimental window, and this difference became more significant with increasing temperature.

### 3.3. Effect of Pt-Coating and Comparison with Disc-Shaped Membranes

As shown in Figure 3, the hydrogen flux was similar under dry sweep with and without Pt-coating, which indicates that Pt coating on the outer surface has little effect on the hydrogen dissociation kinetics on the feed side. However, under wet sweep, the Pt-coating enhanced the apparent hydrogen flux, and the effect becomes more significant as temperature increases, as evidenced from the ratio of hydrogen flux for these two measurements—1.38 at 850 °C vs. 2.23 at 1000 °C. This means that Pt promotes 1) the oxygen exchange kinetics and 2) the charge transfer of the surface reaction due to high electronic conduction of Pt on the outer membrane surface, yielding a larger oxygen gradient between the sweep and the feed sides, and consequently a higher apparent hydrogen flux due to water splitting.

In order to evaluate the hydrogen permeation of LWM-LSC membranes with different thicknesses, architectures, and catalytic coating, the permeability, i.e., flux times membrane thickness for dense ceramic membranes, can be evaluated. In comparison to the disc-shaped membranes reported in [18], the asymmetric membrane in the present work exhibits a lower hydrogen permeability, especially under wet sweep conditions (approximately one order of magnitude lower), as shown in Figure 4. For the Pt-coated disc-shaped membrane, surface kinetics are not expected to limit the hydrogen permeation. Due to the challenge of introducing Pt coating to the inner surface of the asymmetric membranes, i.e., in the region adjacent the membrane/support interface, the feed and sweep sides of the membrane were switched in order to elucidate gas diffusion and/or surface kinetics limitations for the hydrogen permeation and water-splitting process, as described in the following sections.

### 3.4. Effects of Water Splitting

Higher apparent hydrogen permeability in wet as compared to dry sweep gas was observed for all three cercer membranes (symmetric disc shaped, asymmetric tubular, Pt coated asymmetric tubular), and the difference was more significant at higher temperature (c.f. Figure 3 and Figure 4). This has been attributed to water splitting due to transport of oxygen from the sweep to the feed side of the membranes [18,22]. Assuming that the surface kinetics are sufficiently fast, the apparent hydrogen flux due to water splitting caused by ambipolar transport of oxide ions and electrons can be expressed as [25].
(9)$$j_{H_{2}}^{H_{2}O}=\frac{1}{2}j_{O_{2}}=-\frac{RT}{8F^{2}L}\int_{{pO}_{2}^{II}}^{{pO}_{2}^{I}}\sigma_{O^{2-}}t_{e^{-}}d\left(\ln pO_{2}\right)$$
where $\sigma_{O^{2-}}$ is the oxide ion conductivity, and ${pO}_{2}^{I}$ and ${pO}_{2}^{II}$ are the oxygen partial pressure at the feed and permeate side, respectively. For a disc-shaped membrane, a constant pH_2_O at both the feed and sweep sides can be assumed, and given equilibrium between water, hydrogen, and oxygen, Equation (9) can be transformed into
(10)$$j_{H_{2}}^{H_{2}O}=-\frac{RT}{4F^{2}L}\int_{{pH}_{2}^{I}}^{{pH}_{2}^{II}}\sigma_{O^{2-}}t_{e^{-}}d\left(\ln pH_{2}\right)$$

Thus, the measured total hydrogen flux under wet conditions can be expressed as the sum of Equations (9) and (10):(11)$$j_{H_{2}}^{\mathrm{total}}=-\frac{RT}{4F^{2}L}\int_{{pH}_{2}^{I}}^{{pH}_{2}^{II}}\left(\sigma_{O^{2-}}+\sigma_{H^{+}}\right)t_{e^{-}}d\left(\ln pH_{2}\right)$$

LSC shows relatively higher ambipolar conductivity of oxide ions and electron holes than that of LWM, potentially yielding high hydrogen production rate by means of water splitting [16,20,21]. Assuming that the LSC phase in the cercer membrane mainly contributes to the oxide ion conduction and is unvaried across the membrane, the apparent hydrogen flux can be obtained based on Equation (9) using the derived ambipolar conductivity $\sigma_{O^{2-}e^{-}}$ of LSC [20]. As shown in Figure 5, the calculated hydrogen permeability according to the gradients in the two experiments are similar, and much higher than those measured for both the disc and tubular membranes. However, for asymmetric tubular membranes in this work, the measured apparent hydrogen permeability were approximately 2 orders of magnitude lower than the calculated ones. Compared to planar symmetric membranes, asymmetric tubular membranes have an additional porous support and an interface between the support and the thin membrane layer. Therefore, gas diffusion in the porous support and/or surface kinetics at the inner surface could hinder the apparent hydrogen transport. This will be further explored by means of the effect of flow and modeling approaches.

### 3.5. Effects of Flow and Porous Support

Figure 6 presents the effect of sweep flow on the measured hydrogen flux. A higher sweep flow leads to a lower pH_2_ and causes a larger potential gradient for both hydrogen and oxygen transport across the membrane according to Equation (6). This is consistent with the results, i.e., the higher the sweep flow, the higher the hydrogen flux, especially at high temperatures. This effect becomes insignificant above a certain flow, e.g., 100 mL/min at 950 °C, indicating that other limiting factors start to limit the overall transport. Meanwhile, the flow effect becomes weaker as temperature decreases due to the lowered ambipolar transport for both oxygen and hydrogen permeation. 

In order to investigate potential gas diffusion limitations through the porous support, the feed and the sweep sides of the asymmetric membrane were switched, and the apparent hydrogen flux was measured as a function of sweep flow (c.f. Figure 7). 

Under both dry and wet sweep, switching the feed and sweep sides (Figure 8) showed minor differences in the apparent hydrogen fluxes at high temperatures, while a significant increase was observed with outer sweep at low temperatures, especially when using wet sweep gases. Since the outer surface is coated, the catalytic activity for incorporation of oxygen from water splitting into the membrane can be expected to be improved. The increased flux can therefore reasonably be ascribed to the increased kinetics of the Pt-coated membrane surface, while the surface kinetics at the inner membrane/support interface seem to be limiting for oxygen exchange and incorporation.

### 3.6. Numerical Simulation of Gas Transport Across the Membrane

For an asymmetric tubular membrane, gas diffusion in the support causes partial pressure gradients in the porous media, yielding a different potential gradient for gas permeation across the membrane. In this case, Equation (11) is not appropriate for calculating the hydrogen flux since the partial pressure at the membrane/support interface is not defined. Figure 9 shows transport of gas and charged species across the asymmetric membrane under various simplified feed and sweep configurations.

For a steady state gas permeation, we assume in our one-dimensional model that the hydrogen flux is equal across the membrane and support in the radial direction. We can then numerically calculate the partial pressure gradient across the dense membrane layer (region I to II) and the porous support (region II to III) using the DGM explained in Section 2.3. In the case of dry sweep, the ambipolar conductivity of protons and electrons was derived from the flux of the disc-shaped membrane [18] and used to calculate the hydrogen partial pressure at the inner membrane/support interface since the pH_2_ is known at the feed side. Figure 10 shows the pH_2_ distribution along the membrane and the porous support. According to these simulations, there is a significant hydrogen potential gradient across the dense membrane layer, while the pH_2_ inside the support does not vary very much. This indicates that both surface kinetics at the inner membrane/support interface and gas diffusion through the porous support do not limit the overall flux.

In the case of wet sweep, the situation becomes more complex as shown from the schematic illustration in Figure 9b. The steady state flux is again used as an implicit condition. The ambipolar conductivity of oxide ions and electrons was used to derive the oxygen partial pressure at the inner surface of the membrane. The water, oxygen, and hydrogen equilibrium constant was used to derive the partial pressure of oxygen at the feed and the membrane/support interfaces. Figure 11 shows that pH_2_ and pH_2_O are almost uniformly distributed along the porous support, indicating a high performance of the porous support, as also confirmed by previous permeability measurements of the porous support [27]. However, the simulations show a significant drop of approximately 3 orders of magnitude for the pO_2_ at the membrane/support interface (Figure 11b). This implies slow surface kinetics for oxygen incorporation into the cercer composite. Slow kinetics for oxygen incorporation was also evidenced by the change in apparent hydrogen permeation upon switching the feed and sweep side (Figure 8).

As reported earlier, it can be seen from the SEM cross section images in Figure 1 that the membrane/support interface consists mainly of LWM grains and that there are very few LSC grains in the open pores of the support, i.e., in contact with a gas phase. The large difference between the apparent hydrogen flux expected from the simulation results and the measured values may be explained by a barrier to oxygen incorporation into the dense LWM layer at the membrane/support interface. By changing the sweep side, the oxygen incorporation is facilitated to some extent, as more LSC grains are in contact with the gas phase on membrane surface. Thus, an enhanced apparent hydrogen flux (by a factor of about 2) is observed albeit the much slower sweep flow rate. Nevertheless, the apparent flux remains far lower than the theoretically value from Equation 11. Again, the dense LWM layer at membrane/support interface (c.f. Figure 1) seems to be a huge obstacle for oxygen incorporation from steam.

### 3.7. Long-Term Stability

In our previous studies, we have shown that LWM-LSC cercer symmetric membranes degrade under reducing conditions above 900 °C [23]. We also showed that segregation of W may occur on the membrane surface exposed to dry reducing conditions. In the present investigation, the asymmetric membrane was always exposed to hydrogen/oxygen potential gradients, and also dry reducing gradients. Figure 12 shows the reproducible hydrogen fluxes during the first several heating and cooling cycles, and only a small degradation is observed during the last two cooling stages, i.e., from 850–750 °C. This stable performance is quite different from the symmetric membranes [23]. The reason for this difference could be that the interface between the support and the membrane layer prevents cations such as W from diffusing outwards from the surface as observed in [23]. Alternatively, the bulk degradation phenomenon can be expected to be less prominent for the current asymmetric membranes since the apparent flux is limited by surfaces and interfaces rather than ionic conductivity in the bulk. After flux measurements under various conditions, the membrane was kept at 750 °C until the sealing broke. Figure 13 shows stable flux values during the whole period of about 404 h.

## 4. Conclusions

Asymmetric tubular membranes were fabricated by extrusion of an LWM porous support and dip-coating of LWM-LSC. Due to the different sintering characteristics of LWM and LSC, the resulting membranes exhibited a dense LWM layer at the membrane/support interface in addition to the dense LWM-LSC top layer. The permeabilities of the asymmetric membranes were lower than previous measurements of disc-shaped membranes with the same composition, especially under wet sweep conditions. The asymmetric LWM-LSC membrane coated with a porous Pt layer on the outside surface showed comparable hydrogen flux with the uncoated membrane under dry sweep, and higher apparent hydrogen flux with wet sweep on the inside of the tube. By switching the feed and the sweep side, similar hydrogen fluxes were measured under dry sweep, while higher apparent hydrogen fluxes were observed with wet outer sweep, i.e., without significant gas diffusion limitations. Simulations based on a Dusty-gas model and ambipolar transport showed a large drop in pH_2_ and pO_2_ across the membrane/support interface, especially for pO_2_ under wet sweep conditions. The simulations support the experimental findings suggesting that oxygen incorporation at the membrane/support interface limits the apparent hydrogen fluxes especially with wet sweep gas. The membrane showed good stability during various cooling and heating cycles under different feed and sweep conditions, and long-term annealing at 750 °C. The present work highlights critical factors in the fabrication of asymmetric tubular membranes, especially based on composite materials.

## Figures and Tables

**Figure 1 membranes-09-00126-f001:**
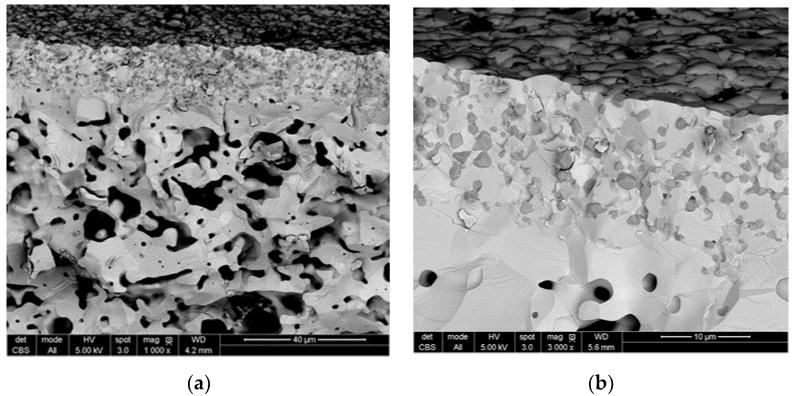
La_27_W_3.5_Mo_1.5_O_55.5−*δ*_–La_0.87_Sr_0.13_CrO_3−*δ*_ (LWM-LSC) membrane sintered at 1500 °C for 10 h observed in cross-section view (fractured surface). The light grey grains are LWM, the dark grains are LSC, and the black areas are pores. Reprint from [27].

**Figure 2 membranes-09-00126-f002:**
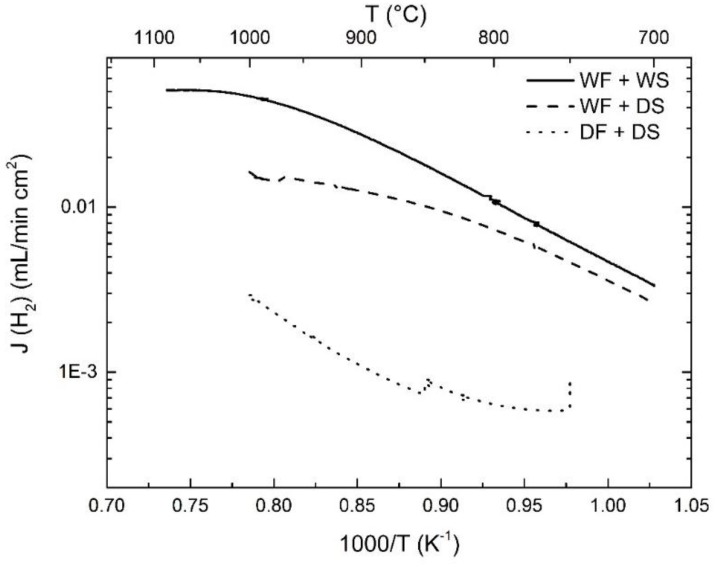
The apparent hydrogen flux as a function of temperature measured upon cooling from high to low temperatures under various feed and sweep gases. The sweep Ar flow is 25 mL/min.

**Figure 3 membranes-09-00126-f003:**
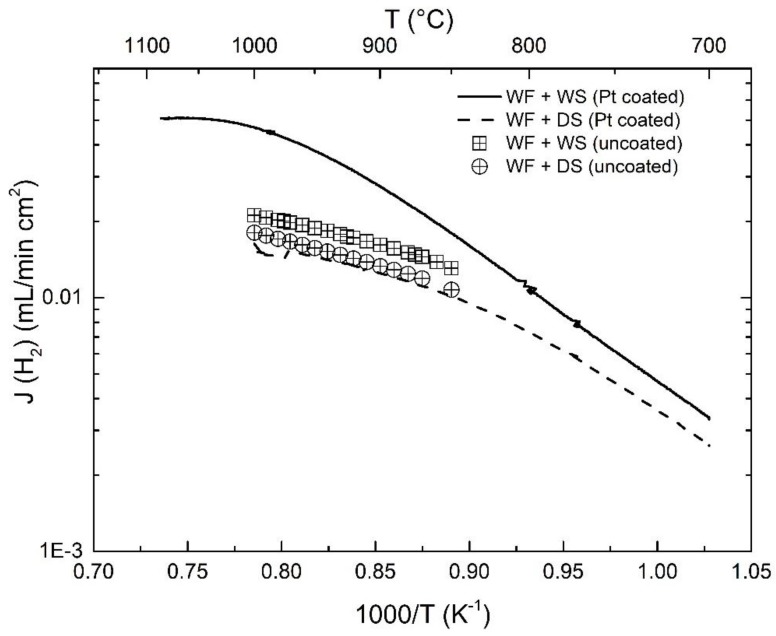
Comparison of hydrogen flux as a function of temperature for Pt-coated and uncoated tubular membranes. The half open labels correspond to the asymmetric tubular membrane without any coating [27]. The sweep Ar flow rate is 25 mL/min.

**Figure 4 membranes-09-00126-f004:**
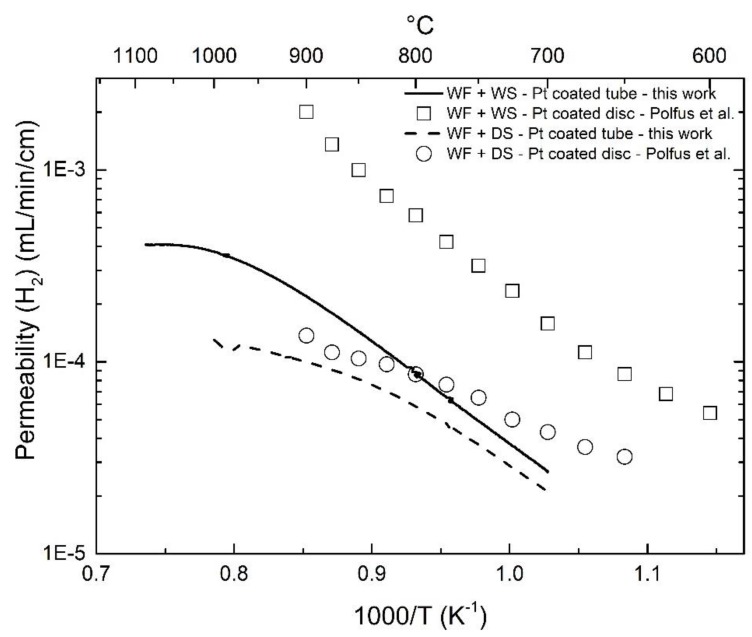
Comparison of hydrogen permeability as a function of temperature for different membrane architectures. The open labels are for disc-shaped membranes coated with Pt at both surfaces [18], and the solid labels for the membrane in this work. The sweep Ar flow is 25 mL/min.

**Figure 5 membranes-09-00126-f005:**
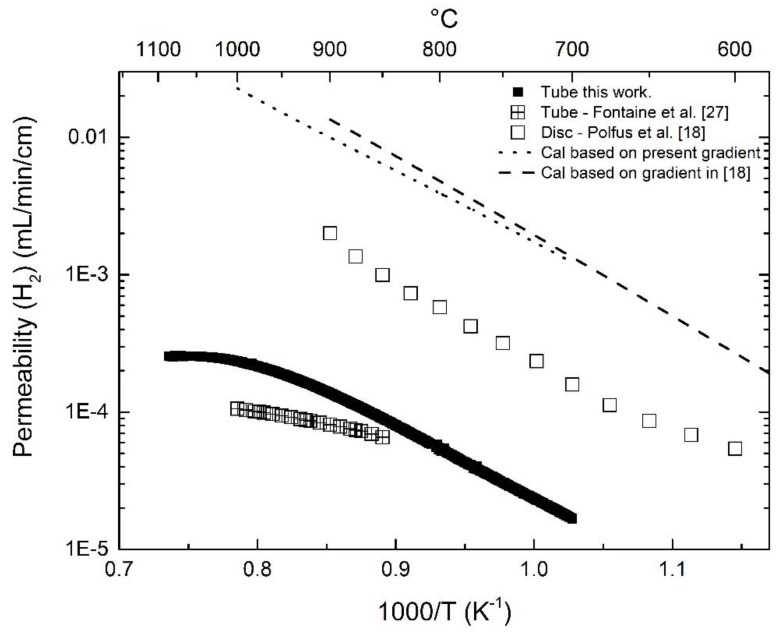
Comparison of hydrogen permeability between experimental values and calculated from the ambipolar conductivity of LSC.

**Figure 6 membranes-09-00126-f006:**
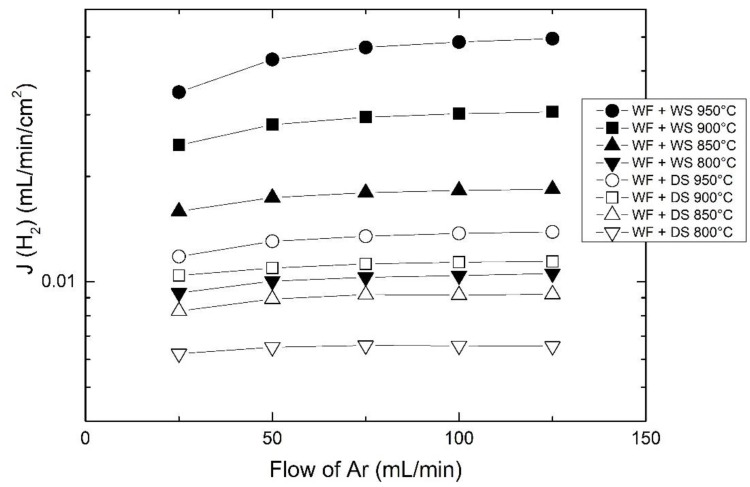
The apparent hydrogen flux as a function of sweep flow in the inner tube. The feed is wet 48.75% H_2_ with a constant flow of 50 mL/min. The temperature ranges from 800 to 950 °C.

**Figure 7 membranes-09-00126-f007:**
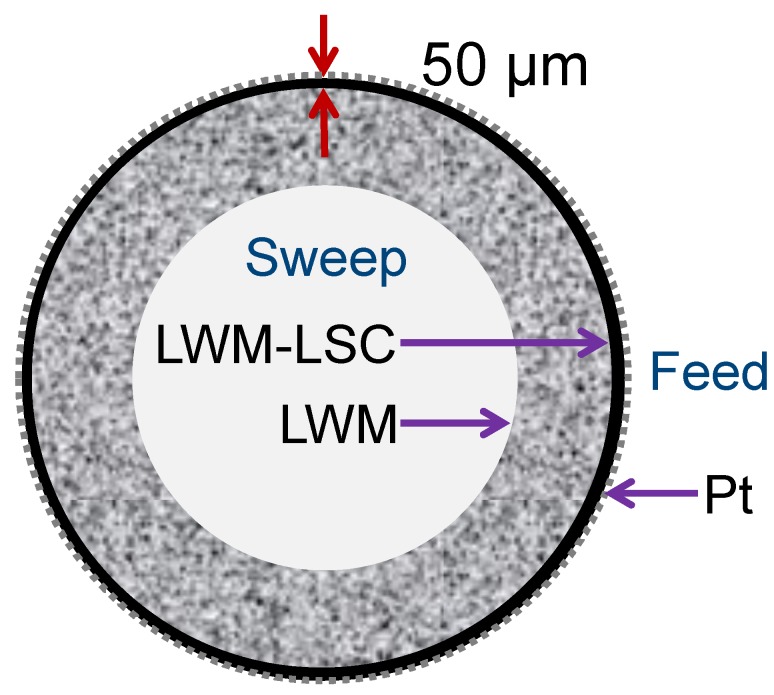
Configuration with inner sweep.

**Figure 8 membranes-09-00126-f008:**
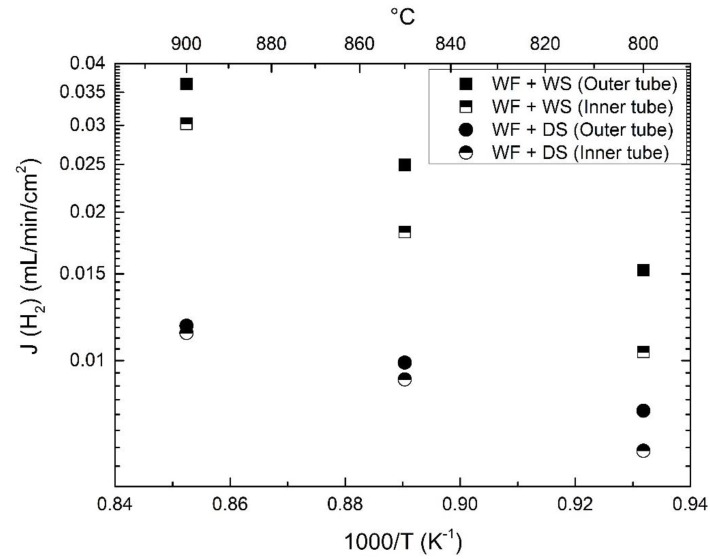
Comparison of sweep side effect on apparent hydrogen flux as a function of temperature. The feed is wet 48.75% H_2_ with a constant flow of 50 mL/min, and the sweep Ar flow is 100 mL/min.

**Figure 9 membranes-09-00126-f009:**
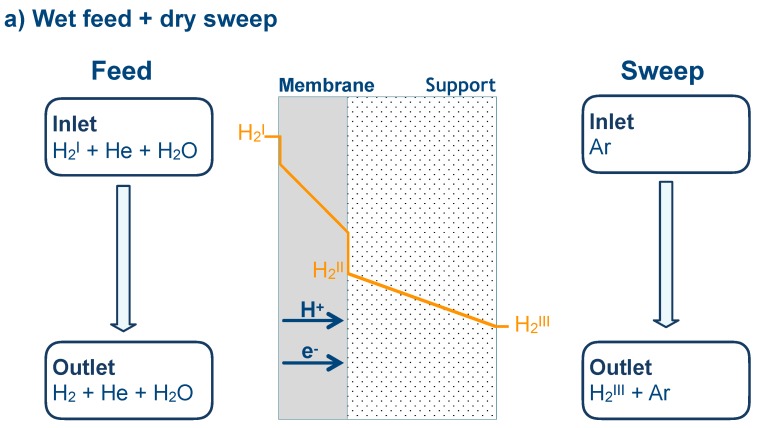
Schematic illustrations of gas transport along asymmetric membranes under various feeds and sweeps. Partial pressure changes along the membrane surface, bulk, and porous support are also visually plotted.

**Figure 10 membranes-09-00126-f010:**
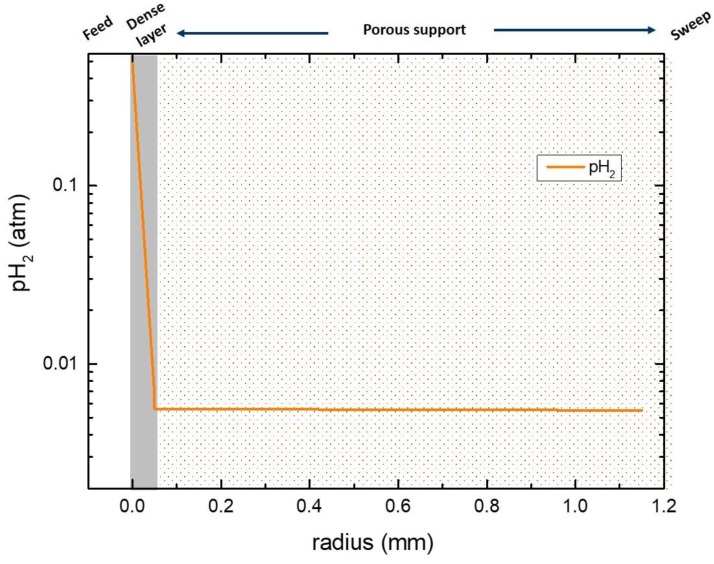
pH_2_ distribution across the membrane layer and the porous support. The sweep is dry Ar, and the temperature is 900 °C.

**Figure 11 membranes-09-00126-f011:**
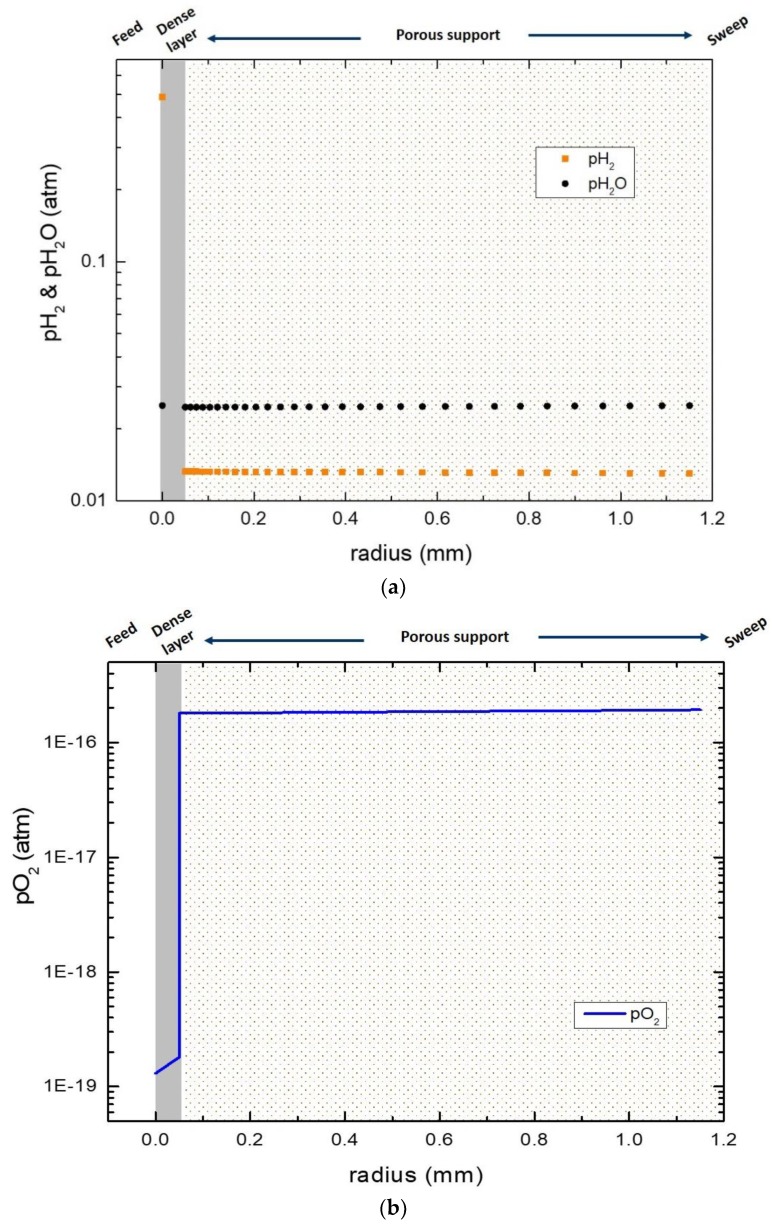
pH_2_, pH_2_O (**a**), and pO_2_ (**b**) distribution across the membrane layer and the porous support. The sweep is wet Ar and the temperature is 900 °C.

**Figure 12 membranes-09-00126-f012:**
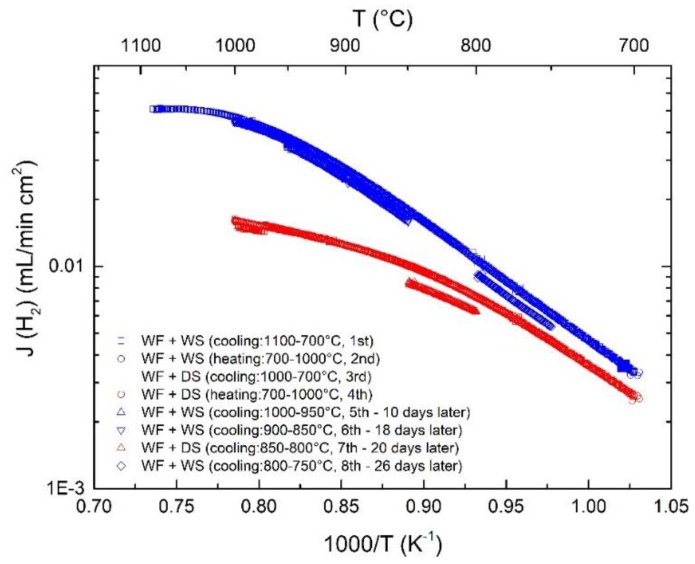
Apparent hydrogen flux as a function of the temperature measured during several heating and cooling cycles using both wet and dry sweeps. The feed side is wet 48.75% H_2_, and the sweep Ar flow is 25 mL/min.

**Figure 13 membranes-09-00126-f013:**
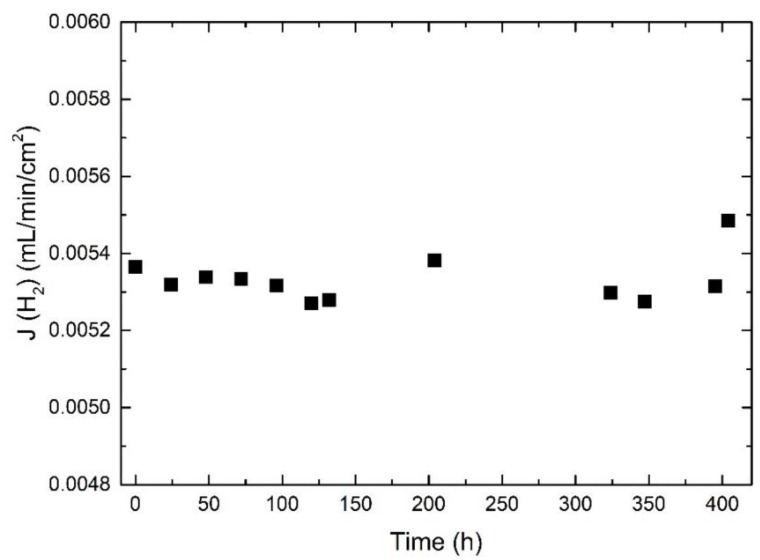
Apparent hydrogen flux as a function of time using WF + WS at 750 °C. The feed side is wet 48.75% H_2_, and the sweep Ar flow is 25 mL/min.

**Table 1 membranes-09-00126-t001:** Parameters of porous support.

Name	Value
Length of the support (mm)	1.1
Porosity	0.46
Tortuosity	3
Pore size (µm)	2

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
