# Peer review of "Factors Limiting the Apparent Hydrogen Flux in Asymmetric Tubular Cercer Membranes Based on La27W3.5Mo1.5O55.5−δ and La0.87Sr0.13CrO3−δ"

_membranes, 2019, doi:10.3390/membranes9100126_

Round 1

Reviewer 1 Report

The authors have presented a well-performed study in the area of processing and characterization of ceramic hydrogen separation membranes. A few comments and recommendations:

Is there a word missing in the title?

Consider defining cercer already in the abstract, as this is a term maybe not familiar to everyone working with membranes

Consider making the novelty of this work compared to previous publications clearer towards the end of the introduction.

Can the diameter of the tube, thickness of the support and the active cer-cer layer be mentioned explicitly somewhere in the text? The same with the porosity, pore size and tortuosity of the support used in the dusty gas model.

Author Response

The authors have presented a well-performed study in the area of processing and characterization of ceramic hydrogen separation membranes. A few comments and recommendations:

Is there a word missing in the title?

Response: This has been fixed.

Consider defining cercer already in the abstract, as this is a term maybe not familiar to everyone working with membranes

Response: We added 'ceramic-ceramic' in the abstract.

Consider making the novelty of this work compared to previous publications clearer towards the end of the introduction.

Response: We modifed the last part of the introduction to make the novelty clearer for the present paper.

Can the diameter of the tube, thickness of the support and the active cer-cer layer be mentioned explicitly somewhere in the text? The same with the porosity, pore size and tortuosity of the support used in the dusty gas model.

Response: All these parameters have been added and listed in Section 2.1 and 3.1 and marked with a red color.

Reviewer 2 Report

The authors have studied well about the limiting factors for the hydrogen and oxygen transport in the asymmetric membranes based on  La27W3.5Mo1.5O55.5-δ–La0.87Sr0.13CrO3 -δ cer-cer composites. The manuscript is well written and the results are convincing. The manuscript may be accepted to "Membranes" journal.

Author Response

The authors have studied well about the limiting factors for the hydrogen and oxygen transport in the asymmetric membranes based on  La27W3.5Mo1.5O55.5-δ–La0.87Sr0.13CrO3 -δ cer-cer composites. The manuscript is well written and the results are convincing. The manuscript may be accepted to "Membranes" journal.

Response: Thanks a lot for the positive feedback.

Reviewer 3 Report

This paper presents an investigation of a ceramic-ceramic membrane to be employed for hydrogen separation. The topic is interesting but the results are presented with almost no discussion or attempt at their interpretation. Major revisions ae required before it is suitable for publication. Specific comments are as follows:

The introduction lacks detail. The authors should add more information regarding the different pros and cons of dense (metallic) and ceramic membranes and the applications for which the use of the proposed membrane is envisaged. The references should be increased to highlight the wealth of works that have already been carried out in the field of hydrogen-permeable membranes. It would be useful to add a picture or schematic representation of the membrane module studied. In Section 3.2, why didn't the authors investigate the effect of temperature on the apparent hydrogen flux whe using dry feed + wet sweep? In section 3.3 (page 5, line 190-195) the authors mention the effect of the Pt coating but no discussion is provided. Why was an effect of Pt on permeation expected? What explanation can be offered for the fact that the Pt coating had an effect on permeation when a wet feed was employed but not in presence of a dry feed? On page 5, lines 202-205,the authors state: "Due to the challenge of introducing a catalyst to the inner surface of the asymmetric membranes, i.e., in the region adjacent the membrane/support interface, the limiting factor for hydrogen permeation in the asymmetric membranes was investigated by varying other experimental parameters." This statement is not clear. In section 3.3 the effect of the design of the membranes on their permeability is discussed, but details are missing. More specifically, no information is provided regarding the Pt loading, membrane surface area, and pressure.  In the section on numerical simulation the equations of the model are missing and no information is provided on the parameters appearing in the model or their values. 

The manuscript would benefit from English editing, starting from the title, which should be modified to "Factors limiting the apparent hydrogen flux in 2 asymmetric tubular cercer membranes based on 3 La27W3.5Mo1.5O55.5-δ and La0.87Sr0.13CrO3-δ" or otherwise corrected.

Author Response

This paper presents an investigation of a ceramic-ceramic membrane to be employed for hydrogen separation. The topic is interesting but the results are presented with almost no discussion or attempt at their interpretation. Major revisions ae required before it is suitable for publication. Specific comments are as follows:The introduction lacks detail. The authors should add more information regarding the different pros and cons of dense (metallic) and ceramic membranes and the applications for which the use of the proposed membrane is envisaged. The references should be increased to highlight the wealth of works that have already been carried out in the field of hydrogen-permeable membranes. It would be useful to add a picture or schematic representation of the membrane module studied.

Response: We agree. We have modified the introduction and added comparisons to Pd-based membranes as well as several other membranes.

In Section 3.2, why didn't the authors investigate the effect of temperature on the apparent hydrogen flux whe using dry feed + wet sweep?

Response: We considered using dry feed+wet sweep. However, we can not determine the exact pO2 when using dry hydrogen in the feed side, therefore we skipped it.

In section 3.3 (page 5, line 190-195) the authors mention the effect of the Pt coating but no discussion is provided. Why was an effect of Pt on permeation expected? What explanation can be offered for the fact that the Pt coating had an effect on permeation when a wet feed was employed but not in presence of a dry feed?

Response: We assume the reviewer meant 'wet sweep'. We have clarified this aspect in Section 3.3: 'This means that Pt promotes 1) oxygen exchange kinetics and 2) charge transfer of surface reaction due to high electronic conduction of Pt on the outer membrane surface, yielding a larger oxygen gradient between the sweep and the feed, and consequently a higher apparent hydrogen flux due to water splitting'.

On page 5, lines 202-205,the authors state: "Due to the challenge of introducing a catalyst to the inner surface of the asymmetric membranes, i.e., in the region adjacent the membrane/support interface, the limiting factor for hydrogen permeation in the asymmetric membranes was investigated by varying other experimental parameters." This statement is not clear.

Response: The text in Section 3.3 has been changed to 'Due to the challenge of introducing Pt coating to the inner surface of the asymmetric membranes, i.e., in the region adjacent the membrane/support interface, the feed and sweep side of the membrane were switched in order to elucidate gas diffusion and/or surface kinetics limitation for the hydrogen permeation and water-splitting process, as described in the following sections'

In section 3.3 the effect of the design of the membranes on their permeability is discussed, but details are missing.

Response: Besides the response above, we also discussed this by means of Wagner transport in Section 3.4: ' Compared to planar symmetric membranes, asymmetric tubular membranes have an additional porous support and an interface between the support and the thin membrane layer'.

More specifically, no information is provided regarding the Pt loading, membrane surface area, and pressure. 

Response: Pt coating usually results in a porous layer of tens of microns. Membrane dimension and total pressure has been added in Section 2.2, marked in red colors.

In the section on numerical simulation the equations of the model are missing and no information is provided on the parameters appearing in the model or their values. 

Response: The dusty gas model described in Section 2.3 has been added. Parameters used for calculations have been added to Section 3.1 and listed in Table 1.

The manuscript would benefit from English editing, starting from the title, which should be modified to "Factors limiting the apparent hydrogen flux in 2 asymmetric tubular cercer membranes based on 3 La27W3.5Mo1.5O55.5-δ and La0.87Sr0.13CrO3-δor otherwise corrected.

Response: The title has been corrected, and we have carefully gone through the whole text and made significant adjustments and improvements throughout the manuscript.

Round 2

Reviewer 1 Report

The small issues I pointed out have now been addressed by the authors.

Reviewer 3 Report

The authors have made appropriate changes